# Altered Gut Microbiota Composition and Its Potential Association in Patients with Advanced Hepatocellular Carcinoma

Ran Huo [1,*,†], Yanlin Chen [2,†], Jie Li [3], Quanguo Xu [4], Junying Guo [1], Haiyan Xu [1], Yiqing You [1], Chaoqiang Zheng [1] and Yan Chen [1,*]

1 Department of Clinical Laboratory, Clinical Oncology School of Fujian Medical University, Fujian Cancer Hospital, Fuzhou 350014, China
2 Department of Clinical Laboratory, Fujian Medical University Union Hospital, Fuzhou 350001, China
3 Department of Clinical Research Center, Dazhou Central Hospital, Dazhou 635000, China
4 School of Pharmacy and Medical Technology, Putian University, Putian 351100, China
* Correspondence: ranhuo863@163.com (R.H.); yanc99@sina.com (Y.C.)
† These authors contributed equally to this work.

**Abstract:** Hepatocellular carcinoma (HCC) is the second-most-common cause of cancer death. In recent years, studies have suggested that intestinal microbiota dysregulation is closely related to HCC and can affect the therapeutic efficacy of immune checkpoint inhibitors. However, there are few data on the relationship between altered gut microbiota composition and its potential association in patients with advanced hepatocellular carcinoma. Hence, in this study, we aimed to investigate the gut microbiota profile associated with advanced hepatocarcinoma. In total, 20 patients with advanced hepatocarcinoma and 20 matched healthy participants were recruited. Stool samples were collected for 16S rRNA sequencing to confirm intestinal microbiota dysbiosis. The results showed that the Nseqs index in advanced hepatocarcinoma patients was significantly different compared with that in healthy individuals, while the butyrate-producing bacteria decreased and LPS-producing bacteria increased. Meanwhile, *Lactobacillus*, *Anaerostipes*, *Fusicatenibacter*, *Bifidobacterium*, and *Faecalibacterium* were significantly correlated with AFP, ALT, AST, and PIVKA. Our findings characterized the gut microbiota composition of advanced hepatocarcinoma, providing an experimental basis and theoretical support for using microbiota to regulate immunotherapy, achieve potential biomarkers for diagnosis, and improve the effect of clinical treatment for patients with advanced hepatocarcinoma.

**Keywords:** advanced hepatocellular carcinoma; gut microbiota dysbiosis; 16S rRNA; gut microbiota biomarkers; immunotherapy

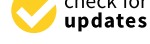



## 1. Introduction

Hepatocellular carcinoma (HCC) is one of the most-common tumor types, and it is generally diagnosed in late stages. In terms of morbidity and mortality, liver cancer ranks as the third highest among various cancers, causing a huge economic burden worldwide [1–3]. Studies have shown that the intestinal microbes in patients with HCC which manifest as more than 30 kinds of intestinal microbes [2], can change in the early stage, indicating that the gut microbiota play an important role in the occurrence and development of HCC [4]. In recent years, studies on the microbe–gut–liver axis have further deepened our understanding of the role of the gut microbiota in promoting the occurrence and development of liver disease. There are active links between liver and intestinal microbes, and their metabolites and products play key roles in the development of HCC. Studies proved that the short-chain fatty acids (SCFAs) produced by intestinal microbiota inhibit deacetylase (HDAC) activity [5], and the reduction in SCFAs can lead to HCC. This decrease in SCFAs is associated with chronic liver disease, which can accelerate the progression of HCC [5–7]. The immunotherapy inhibitors represented by PD-1/PD-L1 are an effective treatment for advanced hepatocarcinoma. Recent studies show that the gut microbiota in tumor patients

can affect the efficacy of immune checkpoint inhibitors (ICBs) [8,9]. Therefore, understanding gut microbiota composition could guide the treatment options and evaluate the efficacy of treatment for patients with advanced hepatocarcinoma. However, in-depth research on the relationship between altered gut microbiota composition and clinical indicators in advanced hepatocarcinoma is limited.

This study intends to investigate the differences in the intestinal microbiota in patients with advanced hepatocarcinoma and healthy people. We wish to provide an experimental basis and theoretical support for using microbiota to regulate immunotherapy, achieve a potential biomarker for diagnosis, and improve the effect of clinical treatment for patients with advanced hepatocarcinoma.

## 2. Materials and Methods

### 2.1. Subjects and Sample Collection

All experimental protocols were reviewed and approved by the Medical Ethics Committee of the cancer hospital affiliated with Fujian Medical University (K2021-118-01), and all individuals signed a written informed consent. Patients with advanced hepatocarcinoma and healthy individuals were enrolled from March 2021 to September 2021 in Clinical Oncology School of Fujian Medical University. The following inclusion and exclusion criteria were adopted.

(1) Inclusion criteria for the healthy group, via the physical examination of individuals in our hospital: (i) age $\geq$ 18 years old; (ii) nonpregnant, lactating, or menstrual women; (iii) no allergic constitution; (iv) no serious primary diseases, such as cardiovascular or cerebrovascular diseases or mental disorders; (v) no underlying diseases and organic diseases; (vi) no antibiotics or probiotics used in the past 2 weeks.

(2) Inclusion criteria for the advanced liver cancer group: (i) age $\geq$ 18 years old; (ii) primary liver cancer diagnosed by pathology; (iii) no allergic constitution; (iv) no participation in other studies within 30 days; (v) no use of antibiotics or probiotics within the past 2 weeks; (vi) no cardiovascular or cerebrovascular diseases or mental disorders.

(3) Exclusion criteria for advanced liver cancer group: (i) antibiotics or probiotics within the past 2 weeks; (ii) secondary liver cancer; (iii) pregnant, lactating, or menstrual women; (iv) allergic constitution; diseases or mental disorders; (v) recent radiotherapy or chemotherapy, cachexia, or cannot tolerate immune checkpoint inhibitor therapy; (vi) immune checkpoint inhibitor therapy; (vii) failure to meet Barcelona Clinic Liver Cancer (BCLC) staging.

Classification standard for advanced hepatocarcinoma group: diagnosed by imaging examinations and histopathological examinations, based on the BCLC staging system, and those who meet the BCLC C stage of the "Primary Liver Cancer Diagnosis and Treatment Standards (2019 Edition)" standard.

In total, 20 patients in the advanced hepatocarcinoma group and 20 participants in the healthy group were enrolled in this study. The clinical characteristics are shown in Table 1. Stool samples were collected. Then, sterile spoons were used to scoop up 2 full spoons (1–3 g) of the stool. The collected feces were placed into the sampling tube and the lid was closed tightly as soon as possible. Finally, we put all in a Ziplock bag, which was sealed, snap-frozen in liquid nitrogen, and stored in −80 °C refrigerator.

### 2.2. 16S rRNA Sequencing and Data Processing

Fecal DNA was extracted by DNeasy PowerSoil Pro Kit (QIAGEN, Germantown, MD, USA), DNA purity and concentration were detected by NanoDrop 2000 (Thermo Fisher Scientific, Waltham, MA, USA), and all extracted DNA samples were frozen in −80 °C refrigerator. The whole process was transported by cold chain with dry ice. V3–V4 region was amplified (ABI, Los Angeles, CA, USA) with upstream primer 338F: ACTCC-TACGGGAGGCAGCAG and downstream primer 806R: GGACTACHVGGGTWTCTAAT. In short, we filtered value that was less than 20 with a tail quality in reads and set a 50 bp window. If the average of quality value was less than 20 in the window, the back-end

bases were truncated from the window, the reads less than 50 bp were filtered after quality control, and the reads containing N bases were removed; the paired reads were spliced into a sequence according to the overlap relationship between the PE reads, and the minimum overlap length was 10 bp; the maximum mismatch ratio of the overlap region in the spliced sequence was allowed to be 0.2, and the non-conforming sequences was screened out; according to the barcodes and primers, the samples were distinguished at the beginning and end of the sequence, and the sequence direction was adjusted; the number of mismatches was allowed to be 0 in the barcode, and the maximum number of primer mismatches was 2 to avoid mismatches.

**Table 1.** Clinical characteristics of patients with advanced hepatocarcinoma (aHCC) and healthy individuals (CON) in this study.

| Characteristics | aHCC (*n* = 20) | CON (*n* = 20) | *p*-Value |
|---|---|---|---|
| Gender, male/female (male %) | 8/12 (40%) | 13/7 (65%) | 0.113 |
| Age, years, median (min–max) | 57.80 (38–73) | 54.80 (51–69) | 0.316 |
| BMI, kg/m$^2$, median (min–max) | 20.62 (15.18–25.84) | 20.74 (14.58–26.38) | 0.914 |
| Stage of clinical characteristics in hepatocellular carcinoma | | | |
| A | / | / | / |
| B | / | / | / |
| C | 20 | / | / |
| Clinical test index, [range], median (min–max) | | | |
| AFP, [0–5.0] | 12,652.6 (1–99,957.0) | 2.3 (1.0–4.0) | 0.000 |
| CEA, [0–7.0] | 3.9 (0.5–12.5) | 2.2 (0.2–4.5) | 0.018 |
| TB, [5.0–21.0] | 20.5 (7.9–43.6) | 15.7 (7.5–24.2) | 0.194 |
| DBIL, [0–8.0] | 5.8 (2.0–18.0) | 3.3 (1.4–6.3) | 0.037 |
| IBIL, [0–20] | 14.7 (5.6–35.9) | 12.4 (4.7–18.3) | 0.394 |
| ALT, [9.0–50] | 66.7 (9.0–248.0) | 20.3 (7.0–36.7) | 0.000 |
| AST, [5.0–40] | 76.5 (17.0–183.0) | 20.7 (12.0–31.6) | 0.000 |
| TP, [65.0–85.0] | 59.5 (31.3–83.1) | 74.6 (68.2–81.2) | 0.000 |
| ALB, [40.0–55.0] | 32.9 (20.1–43.2) | 41.9 (37.6–46.0) | 0.000 |
| PIVKA-II, [0–40.0] | 17,285.5 (12.0–79,000.0) | 18.0 (6.0–36.0) | 0.000 |

The statistical significance of other characteristics was tested by the chi-square test and Wilcoxon rank-sum test; alpha-fetoprotein, AFP; carcinoembryonic antigen, CEA; direct bilirubin, DBIL; alanine transaminase, ALT; aspartate aminotransferase, AST; total protein, TP; albumin, ALB; protein induced by vitamin K absence or antagonist-II, PIVKA-II.

UPARSE software (version 7.1) was used to perform OTU based on 97% similarity. The specific process was as follows: (1) extract non-repetitive sequences from the optimized sequence and remove single sequences without repetition; (2) perform OTU clustering on non-repetitive sequences (excluding single sequences) according to 97% similarity, remove chimeras, and obtain OTU representative sequences during the clustering process; (3) map all optimized sequences to OTU representative sequences, and select sequences with a more than 97% similarity to the OTU representative sequences. Fastp (version 0.20.0) software was used for quality control in the original sequence, and FLASH (version 1.2.7) software was used for splicing.

*2.3. Statistics*

Fastp (version 0.20.0) software was used for quality control in the original sequence of 16S rRNA, and FLASH (version 1.2.7) software was used for splicing. SPSS 10.0 and R software (version 3.3.1) were used for data analysis and graphing, respectively [10–13].

**3. Results**

*3.1. Clinical Characteristics of Advanced Hepatocarcinoma and Healthy Individuals*

To explore whether there are differences in the clinical indicators among these patients, 40 total samples from the HCC group (*n* = 20) and CON group (*n* = 20) were comprehensively analyzed for a bacterial microbiome. No significant difference was observed in age,

gender, or body mass index (BMI) between the two groups (*p* > 0.05). However, there were significant differences in AFP, CEA, DBIL, ALT, AST, TP, ALB, and PIVKA. The AFP, CEA, DBIL, ALT, AST, TP, ALB, and PIVKA in the HCC group were higher compared with those of the CON group (*p* < 0.05, Table 1).

*3.2. α-Diversity and β-Diversity between HCC Group and CON Group*

In our findings, after analyzing 40 stool samples, we found 2,014,511 sequences, including 964,731 sequences in the HCC group, with an average length of 414, and 1,049,780 sequences in the CON group, with an average length of 407, which were divided into 464 OTUs. The good's coverage indices for the observed OTUs in the HCC and CON groups were 99.91% ± 0.028% and 99.92% ± 0.032% (mean ± SD), respectively, demonstrating the sampling reliability. We carried out family-level analysis on all samples and found that the first few communities were *Lachnospiraceae, Ruminococcaceae, Bifidobacteriaceae, Streptococcaceae, Enterobacteriaceae*, and *Peptostreptococcaceae* (Figure 1A), and the proportions of the HCC group were 39.20%, 8.19%, 2.23%, 11.75%, 10.06%, and 3.87%, respectively, while the proportions of the CON group were 52.13%, 16.09%, 10.66%, 1.95%, 1.94%, and 2.66%, respectively. There was a large difference in the colonies of each sample (Figure 1B). Subsequently, Mothur software (http://www.mothur.org/wiki/Calculators) was used to perform α-diversity analysis on the two groups. We found that the Ace index and Heip index were not significant at the OTUs level (Figure 2A,B), while the Nseqs index was significantly different (Figure 2C, *p* < 0.05). In order to analyze the differences in the composition of the gut microbiota between the two groups, the Bray–Curtis distance algorithm was used to perform β-diversity analysis, and PLS-DA analysis found that, at the OTUs level, there were significant differences between the two groups (Amosim, *p* = 0.001; Figure 2D).

*3.3. Changes in Gut Microbes between HCC Group and CON Group*

Afterwards, we analyzed the gut microbes at the phylum and genus levels between the HCC group and the CON group. We found that the abundance of *Actinobacteriota* was higher in the CON group at the phylum level and was as high as *Proteobacteria* and *Patescibacteria* in patients with advanced hepatocarcinoma (Figure 2E). At the genus level, the abundance of *g__Actinomyces, g__Rothia, g__Atopobium, g__Streptococcus, o__Coriobacteriales*, and *g__Scardovia* in patients with advanced hepatocarcinoma was significantly higher (Figure 2F).

In order to explore the relationship between the HCC group and the CON group, the nonparametric Kruskal–Wallis (KW) sum-rank test was used to detect the differences in the species abundance. Then, the Wilcoxon rank-sum test was used to detect the consistency of the different species in the subgroups. Finally, LDA analysis was used to estimate the influence of the abundance of each component on the differential effect, and we found that the abundance of *o__Lactobacillales, c__Bacilli, p__Proteobacteria, c__Gammaproteobacteria, f__Enterobacteriaceae, o__Enterobacterales, f__Streptococcaceae, g__Streptococcus, g__Klebsiella, f__Lactobacillaceae, g__Lactobacillus, f__Prevotellaceae, g__Prevotella, g__Scardovia, g__Sutterella, g__norank_f__Peptococcaceae, f__Erysipelotrichaceae, o__Pasteurellales, f__Pasteurellaceae, g__Haemophilus, g__Atopobium, g__Anaerofustis, f__Anaerofustaceae, g__Megasphaera*, and *g__Alloscardovia* in the HCC group and *c__Clostridia, f__Lachnospiraceae, o__Lachnospirales, o__Oscillospirales, f__Ruminococcaceae, g__Bifidobacterium, o__Bifidobacteriales, f__Bifidobacteriaceae, c__Actinobacteria, p__Actinobacteriota, g__Faecalibacterium, g__Blautia, g__Fusicatenibacter, g__Eubacterium_hallii_group, f__Erysipelatoclostridiaceae, g__Erysipelotrichaceae_UCG-003, g__Agathobacter, g__Anaerostipes, g__Anaerococcus, f__Phormidiaceae, g__Arthrospira, o__Cyanobacteriales, o__Monoglobales, f__Monoglobaceae, g__Monoglobus, g__Candidatus_Stoquefichus, f__Butyricicoccaceae, g__Butyricicoccus, f__Sutterellaceae, g__Frisingicoccus, g__Tyzzerella, g__Oscillibacter, g__Paraprevotella*, and *g__Eubacterium_ventriosum_group* in the CON group had a great influence on the abundance of bacteria between the two groups by analysis from the phylum level to the genus level (LDA > 3, *p* < 0.05, Figure 3).

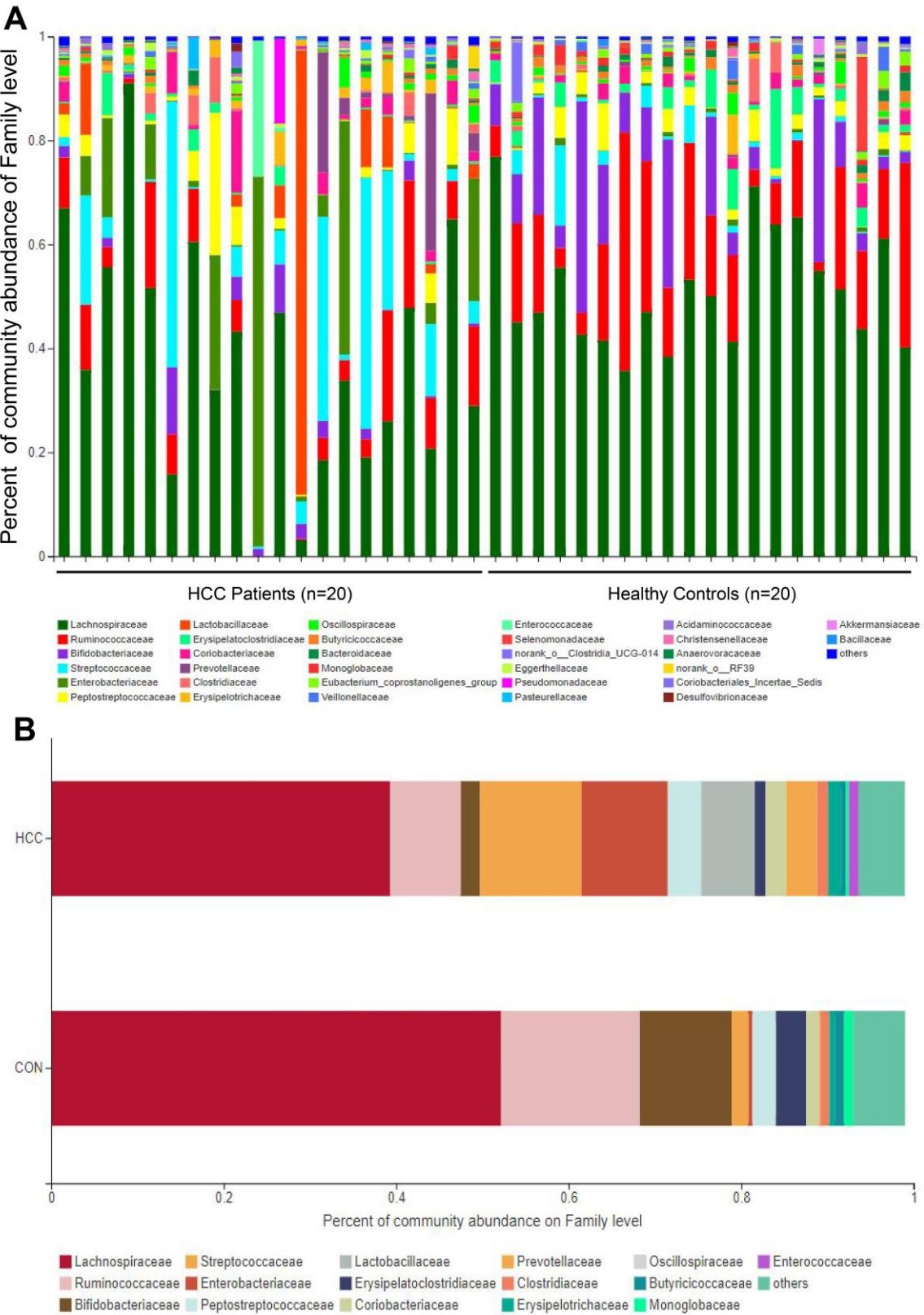

**Figure 1.** (**A**) Relative abundance of gut microbiota at family level of each sample, with different colored columns representing different species and the length of the column representing the relative abundance of the species. (**B**) Relative gut microbiota abundance at family level in HCC and CON groups.

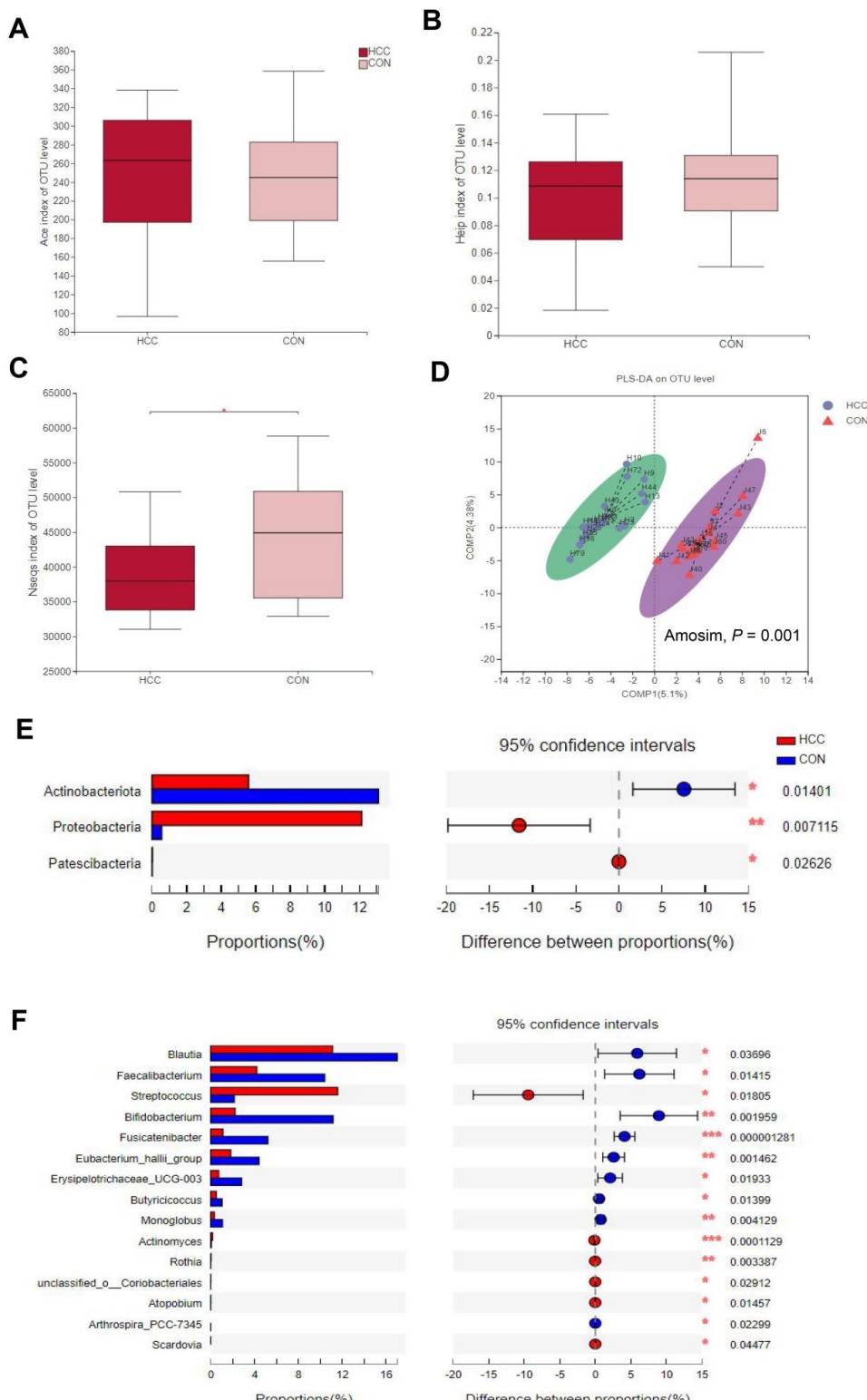

**Figure 2.** (**A**) Ace index, (**B**) Heip index, and (**C**) Nseqs index: differences between HCC group and CON group. (**D**) Partial least-squares discriminant analysis (PLS-DA) for OTUs between HCC group and CON group (ANOSIM, $R = 0.28$, $p = 0.001$). Wilcoxon rank-sum test. (**E**,**F**) Differences in gut microbiota at different levels between HCC group and CON group. (**E**) HCC group compared with CON group at phylum level. (**F**) Comparison of gut microbes at genus level between groups; * $p < 0.05$, ** $p < 0.01$, *** $p < 0.001$. All tested by Student's *t*-test.

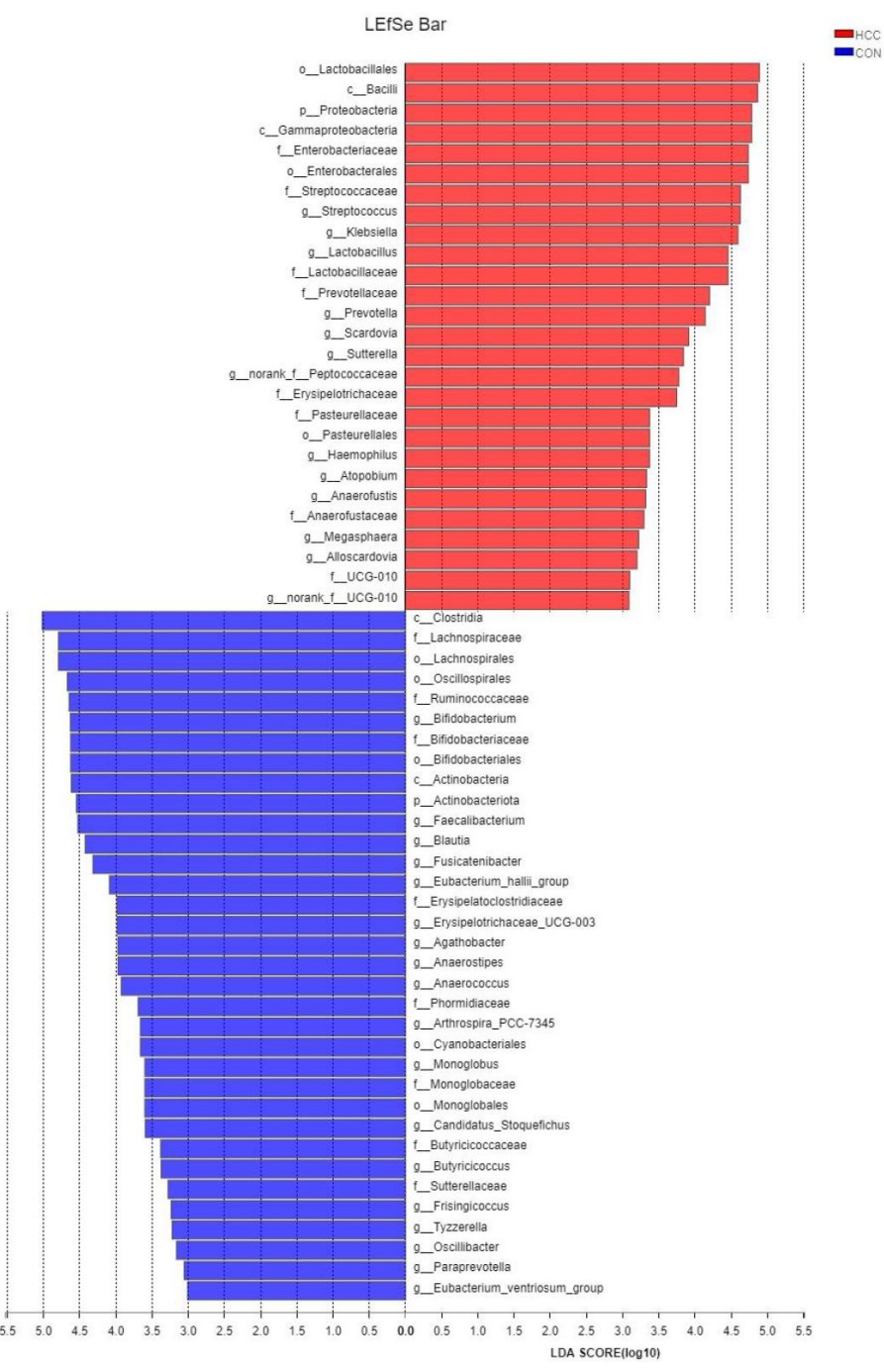

**Figure 3.** LDA discriminant histogram shows differences detected between HCC group and CON group, with LDA value > 3, and one-against-all was adopted to compare two groups.

*3.4. Correlation Analysis between Clinical Indicators and Gut Microbiota*

We correlated the clinical indicators collected from these patients, including AFP, CEA, TB, DB, IB, ALT, AST, TP, ALB, and PIVKA, with the top 20 bacterial species in the intestinal abundance and the preceding heatmap analysis, we found that *Coprococcus, Erysipelotrichaceae_UCG-003, Dorea, unclassified_f__Lachnospiraceae, Ruminococcus, Lactobacillus, Klebsiella, Anaerostipes, Fusicatenibacter, Eubacterium_hallii_group, Agathobacte, Bifidobacterium,* and *Faecalibacterium* were correlated with these clinical indicators at the genus level, among which *Lactobacillus, Anaerostipes, Fusicatenibacter, Bifidobacterium,* and *Faecalibacterium* had a high correlation with the clinical indicators, as shown in the heatmap (* $p < 0.05$, ** $p < 0.01$, *** $p < 0.001$, Figure 4A).

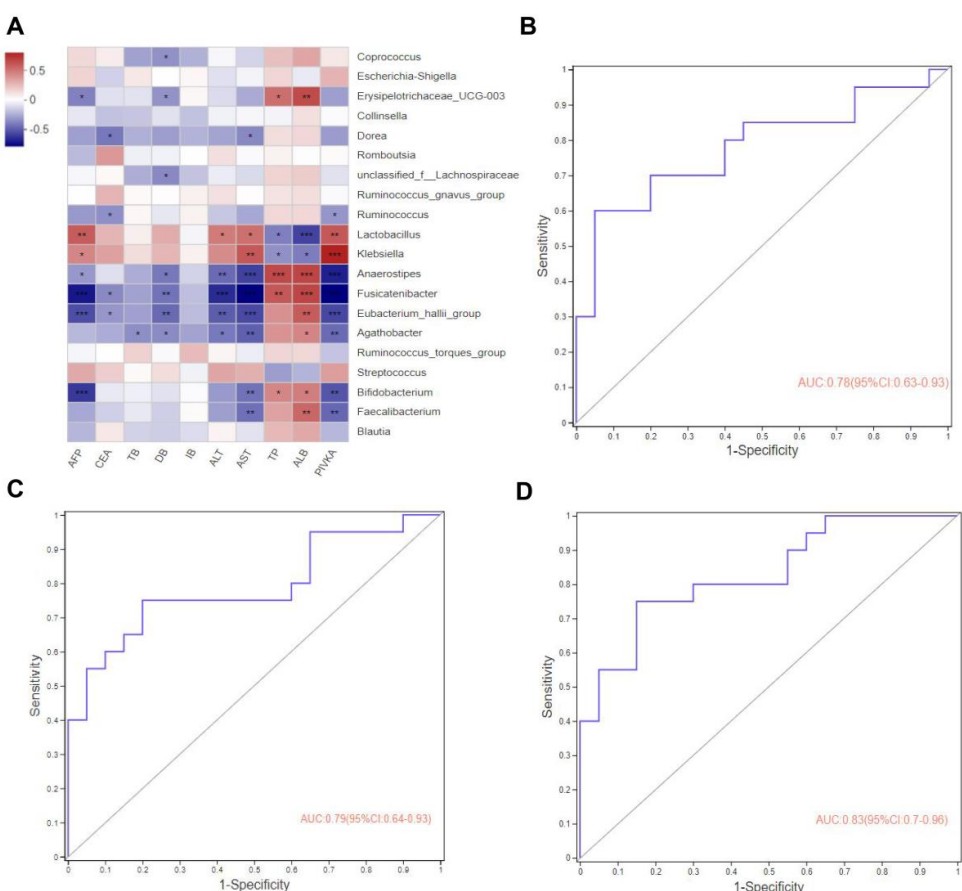

**Figure 4.** (**A**) The correlation test of the top 20 intestinal microbiota abundances and clinical indicators between HCC group and CON group. Clinical indicators and species are presented as the X-axis and Y-axis, respectively, and the *R* values and *p* values were obtained by calculation. *R* values are shown as different colors in the figure, and the color interval of different *R* values is presented as the legend on the right. All data were analyzed by Spearman's correlation. (**B–D**) ROC diagnostic curve of gut microbes. (**B**) ROC curve of the top 4 intestinal microbiota in abundance. (**C**) ROC curve of the top 7 intestinal microbiota in abundance. (**D**) ROC curve of the top 10 intestinal microbiota in abundance. * $p < 0.05$, ** $p < 0.01$, *** $p < 0.001$.

### 3.5. Gut Microbes May Predict Potential Biomarkers for Advanced Hepatocarcinoma

Subsequently, based on the LEfSe, a stepwise logistic regression model and a random forest model were used as feature predictors at the genus level. We compared the top 4 (*Fusicatenibacter*, *Anaerostipes*, *Lactobacillus*, and *Roseburia*), top 7 (*Fusicatenibacter*, *Anaerostipes*, *Lactobacillus*, *Roseburia*, *Monoglobus*, *Eubacterium_hallii_group*, and *Tyzzerella*) and top 10 (*Fusicatenibacter*, *Anaerostipes*, *Lactobacillus*, *Roseburia*, *Monoglobus*, *Eubacterium_hallii_group*, *Tyzzerella*, *Arthrospira_PCC-7345*, *Eubacterium_eligens_group*, and *Erysipelotrichaceae_UCG-003*) using the receiver operating characteristic curve (ROC). We found that the area under the curve (AUC) was 0.78 (95% CI: 0.63–0.93), 0.79 (95% CI: 0.64–0.93), and 0.83 (95% CI: 0.70–0.96), respectively, indicating that these gut microbiota have a certain diagnostic value (Figure 4B–D).

### 3.6. Enzyme Functions and Enzyme–Metabolite Predictions on Gut Microbiota

In order to understand enzyme functions and enzyme differences in gut microbiota, firstly, we standardized the OTU abundance table by PICRUSt to remove the influence of the 16S marker gene in the genome of the species; then, we obtained the COG family information and KEGG Ortholog (KO) information through the corresponding Gene ID. Each COG abundance and KO abundance was calculated. According to the KEGG database

information, the KO, pathway, and EC information were obtained, and the abundance of each functional category was calculated according to the OTUs. In addition, PICRUSt was used to obtain three levels of metabolic pathway information and obtain the abundance table of each level. We predict that the changes in gut microbiota may mainly affect the xenobiotics' biodegradation and metabolism, metabolism of other amino acids, biosynthesis of other secondary metabolites, lipid metabolism, and amino acid metabolism (Figure 5).

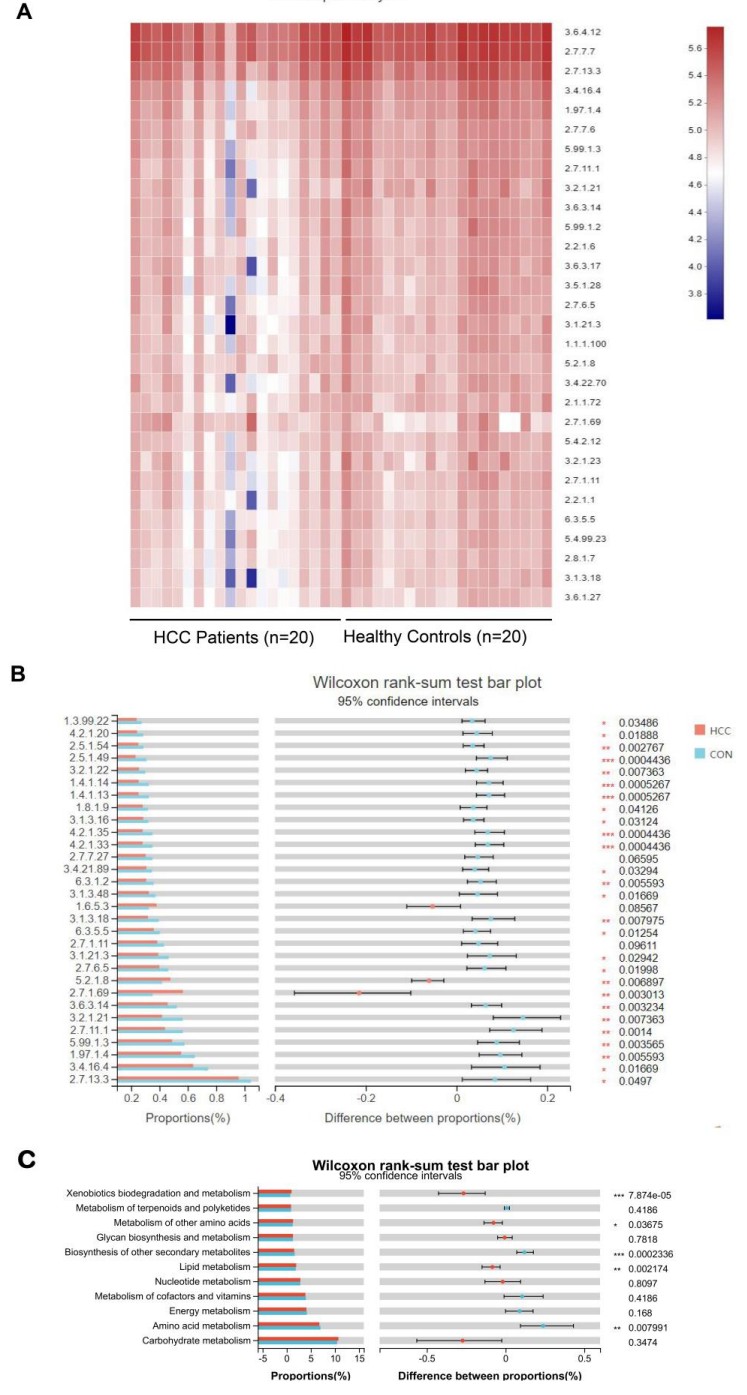

**Figure 5.** (**A**) HCC group and CON group are presented as X axis, and KEGG enzyme number is presented as Y axis. The color gradient of the color block is used to display the changes in the different functions between groups. The legend is the value represented by the color gradient. (**B**,**C**) Enzyme abundance and differences between HCC group and CON group. * $p < 0.05$, ** $p < 0.01$, *** $p < 0.001$. Wilcoxon rank-sum test was used.

## 4. Discussion

In summary, the types and functions of gut microbiota in the human intestine are highly diverse, including bacteria, viruses, fungi, and archaea. The total number of gut microbiota is more than $1 \times 10^{14}$, which is 10 times the number of human cells [14]. In recent years, gut microbiota have become an international hotspot in academic research. The fact that changes in the intestinal microbiota can influence the efficacy of immune checkpoint inhibitors (ICBs) in patients with advanced hepatocarcinoma has attracted much attention from scholars around the globe. Matthias Pinter et al. discussed that the survival of patients with advanced hepatocarcinoma could be greatly improved by ICBs, expounded the prospects and challenges to ICBs, and pointed out the need for unsatisfied biomarkers to directly predict therapeutic response or drug resistance [15]. However, in previous studies [16,17], researchers did not correlate the clinical indicators with the gut microbes in patients with HCC. Thus, we compared the differences in the gut microbes in patients with advanced hepatocarcinoma in this study, and the changes in the gut microbes with clinical indicators combined with the analysis of diagnostic specificity, which play a role in the diagnosis of HCC, provide some diagnostic biomarkers for the clinical diagnosis and improve ICBs' treatment.

In our design, the 16S rRNA analysis for patients with advanced hepatocarcinoma without clinical intervention and healthy individuals significantly revealed the changes in the intestinal microbiota. We found that the abundance of *p__Proteobacteria*, *p__Patescibacteria*, *g__Actinomyces*, *g__Rothia*, *g__Atopobium*, *g__Streptococcus*, *g__unclassified*, *o__Coriobacteriales*, and *g__Scardovia* increased, and *Proteobacteria*, *Streptococcus*, and *Bifidobacterium* were basically consistent with previous reports [17]. Some contradictory results may be due to various factors, including the CON group, tumor status, local diet, and other factors, which may influence certain bacterial communities. In some diseases, decreased microbial diversity was considered to be one of the main features of intestinal microbial dysbiosis. In this study, there was no difference in the Ace index and Heip index compared with the CON group, which was inconsistent with the reports [16–18]. Although we found no significant difference in the Ace index and Heip index of the gut microbiota between the advanced HCC group and the healthy CON group, we found significant differences in the Nseqs index in the advanced hepatocarcinoma patients, and the HCC group showed a significant decrease in the number of sequences. The Heip index represents the uniformity of the sample community, while Nseqs represents the number of sequences. We believe that these two values represent the different levels of diversity, so it does not mean that all the exponential changes are consistent. In addition, PLS-DA analysis showed that there were significant differences between the two groups, indicating that there was a certain disorder in the intestinal microbiota between the two groups. Moreover, the changes in the intestinal microbiota were closely related to dietary factors [19,20], so the results might be inconsistent with the dietary structure reported by Behary J et al. [18]. Therefore, the dietary habits of the subjects could be different from those in previous studies. We hypothesized that the gut microbiota in patients can be influenced by the area or the environment.

In addition to the changes in richness and diversity, we found that *Proteobacteria* was increased in patients with advanced hepatocarcinoma, which was consistent with the report by Liu, Q. et al. [21]. We speculated that *Proteobacteria* could be a biomarker to indicate microbial dysbiosis [22,23], and the overgrowth of these bacteria might be associated with a high-fat diet. In this study, it was found that there was not a significant difference in the BMI indexes of advanced hepatocarcinoma patients and healthy individuals. We speculated that the underlying hepatic pathological state might lead to the metabolic disorders in the body, resulting in the changes in *Proteobacteria*. It is also possible that a high-fat diet may act as a risk factor for HCC development. Subsequently, after analyzing the two groups at the genus level, we found that the abundance of *Bifidobacterium* was higher than the healthy group, which was consistent with a previous report [18]. *Bifidobacterium* was found to be significantly reduced in the patients with HCC, and the *Bifidobacterium/Enterobacteriaceae* ratio can be used as an indicator of hepatic disease progression for biological balance [24]. *Bifidobacterium*

and *Bacteroides* were proven to be functioning in bile saline hydrolysis, which could convert intestinal bound bile acids into unbound bile acids, playing a metabolic-regulation role through bile acid-related signaling pathways [25]. Furthermore, Sivan A et al. found that *Bifidobacterium* enhanced the efficacy of immunotherapy for malignant tumors, guide immunotherapy, or the prognosis of treatment [8].

After using LEfSe multi-level species analysis, we found that the butyrate-producing bacteria in the CON group, including *Clostridia*, *Lachnospirales*, *Oscillibacter*, *Faecalibacterium*, *Ruminococcaceae*, and *Bifidobacteriaceae*, significantly increased compared with those in patients with HCC. However, LPS-producing bacteria such as *Klebsiella* and *Haemophilus* increased in patients with advanced hepatocarcinoma, which was consistent with reports from Hangzhou, China [2]. We speculated that butyrate, as the main energy source in the intestinal mucosa, was considered to be an important regulator of cell gene expression in the host, inflammation, differentiation, and apoptosis and played a key role in bacterial energy metabolism and intestinal health. Therefore, the reduction in butyrate-producing bacteria could promote intestinal mucosa destruction and the development of HCC [26]. Nevertheless, LPS could trigger various pathophysiological cascades [27]. The *NF-κB* pathway was activated by high-level LPS and produced proinflammatory cytokines (TNF-$\alpha$, IL-6, and IL-1), leading to inflammation and oxidative damage in the liver [28]. These data suggested that altering the gut microbiota could interfere with potential biomarkers in HCC progression through the gut–microbiota–liver axis.

Clinically, an ultrasound, combined with AFP in serum, is used to monitor early HCC, but AFP is also elevated in embryonic-derived tumors, acute and chronic hepatitis, and other diseases. There are some defects in the non-specific diagnostic indexes such as ALT and other diagnostic enzymes, and these may result in a high false-negative or false-positive rate as diagnostic biomarkers [29]. There may be a correlation between the diagnostic indicators and the changes in the intestinal microbiota in patients with HCC. Then, we conducted a Spearman rank correlation test for the top 20 microbiota changed in the genus level between the clinical indicators and intestinal microbiota. It was found that *Lactobacillus*, *Anaerostipes*, *Fusicatenibacter*, *Bifidobacterium*, and *Faecalibacterium* were significantly correlated with AFP, ALT, AST, and PIVKA. These results suggested that there was a direct or indirect relationship between these clinical indicators and the intestinal microbiota in advanced hepatocarcinoma. In view of the above, we analyzed the top 4, 7, and 10 intestinal microbiota (*Fusicatenibacter*, *Anaerostipes*, *Lactobacillus*, *Roseburia*, *Monoglobus*, *Eubacterium_hallii*, *Tyzzerella*, *Arthrospira_PSC-7345*, *Eubacterium_eligens_group*, and *Erysipelotrichaceae_UCG-003*) at the genus level, which were found to have a certain diagnostic effect via the ROC diagnostic curve, suggesting that these microbiota could have a certain value in the diagnosis of advanced hepatocarcinoma. Zmora N. et al. also found that *Fusicatenibacter* produced butyrate, an SCFAs-producing bacterium, and its abundance dynamically changes with the progress of the disease [30]. The changes in *Anaerostipes*, *Lactobacillus*, and others as probiotics in patients with HCC are consistent with a previous report [17]. These studies revealed that these microbiota could provide some theoretical support for the clinical evaluation and treatment of advanced hepatocarcinoma. Then, the KEGG database was used for the heatmap via a PICRUSt2 prediction combined with the KEGG website. It was also found that the enzymes in the intestinal microbiota were basically energy metabolism enzymes and function metabolism enzymes, including xenobiotics' biodegradation metabolism and amino acids' metabolism. Studies found that the changes in the intestinal microbial enzymes in patients with HCC were closely related to the progress of the disease and to various metabolic enzymes, which was the same as shown in previous reports [31–34]. However, these data are only a prediction and will need to be confirmed further. Combined with the results above, we speculate that the intestinal microbiota is changed in patients with advanced hepatocarcinoma.

## 5. Conclusions

This study revealed the differences in the intestinal microbiota between patients with advanced hepatocarcinoma and healthy individuals and analyzed the connection between the intestinal microbiota and the clinical indicators in patients, which found some potential biomarkers in the intestinal microbiota for diagnosis and provided immunotherapy treatment combined with microbiology for advanced hepatocarcinoma. However, this has certain limitations. First of all, because patients with advanced hepatocarcinoma have generally been on treatment, it is difficult to collect specimens from additional diagnosed patients, so the sample size is not enough. Furthermore, this research on the gut microbiota was analyzed in Fujian, China, and lacked patients from multiple research centers or regions. Finally, we will use different bacteria we have obtained for animal experiments, to clarify the potential bacteria that improve the efficacy of immunotherapy, the effect of early diagnosis, and the clinical treatment of patients with advanced hepatocarcinoma.

**Author Contributions:** R.H., Y.C. (Yan Chen), Y.C. (Yanlin Chen), J.L., Q.X., J.G., H.X., Y.Y. and C.Z.: designed and performed the experiments; R.H. and J.L.: wrote the manuscript. All authors have read and agreed to the published version of the manuscript.

**Funding:** This work was supported by the National Natural Science Foundation of China (Youth Program, 82102468) and the Natural Science Foundation of Fujian Province (General Program, 2022J011043).

**Institutional Review Board Statement:** The study was conducted in accordance with the Declaration of Helsinki and was approved by the Medical Ethics Committee of the cancer hospital affiliated with Fujian Medical University (K2021-118-01, December 2021).

**Informed Consent Statement:** Informed consent was obtained from all subjects involved in the study.

**Data Availability Statement:** The data presented in this study are available on request from the corresponding author.

**Conflicts of Interest:** The authors declare no conflict of interest.

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
