# Peer review of "Altered Gut Microbiota Composition and Its Potential Association in Patients with Advanced Hepatocellular Carcinoma"

_curroncol, doi:10.3390/curroncol30020141_

Round 1
Reviewer 1 Report
General comments:
The study Altered gut microbiota composition and its potential association in patients with advanced hepatocellular carcinoma gives insight into the differences between microbiota composition in hepatocellular carcinoma patients and healthy participants. The novelty of this study is not properly highlighted. The scientific background and deeper analysis of this problem should be described, such as the rationale for the study. The literature review is not strong enough to provide research gaps for this research work, so introduction section should be revised to give latest literature review according the topic. The English language used in the manuscript needs major improvements as there are many punctuation and grammatical mistakes present throughout the manuscript. Sentences need more clarity and better construction. It is obvious the quality of the manuscript does not meet the standards of Current Oncology Journal, therefore needs major revisions or should be rejected in its present form.
Specific comments:
1. Authors are advised to revise keywords section, and add some more specific keywords.
2. Page 2, section 2.2. S rRNA sequencing 70- references should be add for each standard method.
3. Authors are advised to revise all figures, because the given one are poor quality with unreadable parts.
4. Authors are advised to revise and update references, as the given one are not enough to support the study.
5. Authors are advised to discuss the study results with the latest one, as the given references are not strong enough to present the credibility of results.
6. Authors are advised to the conclusion section in the following way: Key values from results; Major findings and contribution; As well as limitations of the study if there is any; Possibile future work; Make it up to 250-300 words.
Author Response
General comments:
The study Altered gut microbiota composition and its potential association in patients with advanced hepatocellular carcinoma gives insight into the differences between microbiota composition in hepatocellular carcinoma patients and healthy participants. The novelty of this study is not properly highlighted. The scientific background and deeper analysis of this problem should be described, such as the rationale for the study. The literature review is not strong enough to provide research gaps for this research work, so introduction section should be revised to give latest literature review according the topic. The English language used in the manuscript needs major improvements as there are many punctuation and grammatical mistakes present throughout the manuscript. Sentences need more clarity and better construction. It is obvious the quality of the manuscript does not meet the standards of Current Oncology Journal, therefore needs major revisions or should be rejected in its present form.
Specific comments:
- Authors are advised to revise keywords section, and add some more specific keywords.
A:Thank you for your suggestion. We have some more specific keywords, such as gut microbiota biomarkers and immunotherapy.
- Page 2, section 2.2. S rRNA sequencing 70- references should be add for each standard method.
A: We have added references as follows:
Liu C, Zhao D, Ma W, et al. Denitrifying sulfide removal process on high-salinity wastewaters in the presence of Halomonas sp[J]. Applied microbiology and biotechnology, 2016, 100(3): 1421-1426. doi:10.1007/s00253-015-7039-6,
Chen S , Zhou Y , Chen Y , et al. fastp: an ultra-fast all-in-one FASTQ preprocessor[J]. Bioinformatics, 2018, 34(17):i884-i890. doi:10.1093/bioinformatics/bty560, Tanja, Mago, Steven, et al. FLASH: fast length adjustment of short reads to improve genome assemblies.[J]. Bioinformatics, 2011, 27(21):2957‐2963. doi:10.1093/bioinformatics/btr507,
Edgar, Robert C . UPARSE: highly accurate OTU sequences from microbial amplicon reads.[J]. Nature Methods, 2013, 10(10):996‐998. doi:10.1038/nmeth.2604 ,Wang Q . Naive Bayesian classifier for rapid assignment of rRNA sequences into the new bacterial taxonomy[J]. Appl. Environ. Microbiol, 2007, 73. doi:10.1128/AEM.00062-07
- Authors are advised to revise all figures, because the given one are poor quality with unreadable parts.
A:Your suggestion is very good. We have improved the resolution of all Figures.
- Authors are advised to revise and update references, as the given one are not enough to support the study.
A:Thank you for your suggestion. Since there are few on our research on intestinal microbes and clinical indicators of advanced liver cancer, we try our best to search relevant update references.
- Authors are advised to discuss the study results with the latest one, as the given references are not strong enough to present the credibility of results.
A:Due to the limitation of space, references we quoted are high scores,and strongly support our views,According to your suggestion,we have added references.
- Authors are advised to the conclusion section in the following way: Key values from results; Major findings and contribution; As well as limitations of the study if there is any; Possibile future work; Make it up to 250-300 words.
A:Thank you for your suggestion. We have adjusted the structure of the manuscript and added conclusion section “This study has found that there are differences in intestinal microbiota between patients with advanced hepatocarcinoma and healthy individuals, and analyzed the connection between the intestinal microbiota and the patient's clinical indicators, found that some potential biomarkers on intestinal microbiota for diagnosing, and provide the treatment of immunotherapy combined with microbiology for advanced hepatocarcinoma. However, it has certain limitations. First of all, because patients with advanced hepatocarcinoma have been taken measures on treatment generally, it is difficult to collect specimens from more diagnosed patients, so the sample size is not enough. Furthermore, this research on gut microbes were analyzed in Fujian, China, and the lack of the patients among multi-research centers or regions. Finally, we will further use different bacteria we have obtained for animal experiments, to clarify the potential bacteria that improve the efficacy of immunotherapy and the effect of early diagnosis and clinical treatment of patients with advanced hepatocarcinoma.”
Reviewer 2 Report
Hepatocellular carcinoma (HCC) is one of the most common causes of cancer death. The association between gut microbiota and various stage of HCC has been largely investigated1–3. In this work, the authors examined the differences and internal relations of gut microbiota between advanced HCC patients and healthy people.They showed Nseqs index in advanced HCC patients was significantly different compared with healthy individuals, with butyrate-producing bacteria decreased and LPS producing bacteria increased, which is consistent with previous research2. Meanwhile, they also found Lactobacillus, Anaerostipes, Fusicatenibacter, Bifidobacterium and Faecalibacterium were significantly correlated with AFP, ALT, AST and PIVKA. The research is straight-forward but the findings are not very novel, I recommend that this paper not be accepted without major revision.
There are several major and minor points that should be addressed:
Major:
1. The authors should explain more in the introduction about why they focused on the differences in gut microbiota between advanced (not other stages) HCC patients and healthy people. There is already work done in this1. What is the novelty of the work?
2. For figure 5, there is no detailed description in the main text about the results analyzed by PICRUSt. The paper could benefit from an explanation for enzyme function and explanation for differences in enzyme abundance between patients with disease and healthy people. More explanation about Fig. 5 should be added to the main text.
3. In figure 2, the authors found that the Ace index and Heip index were not significant at the OTUs level, while the Nseqs index were significantly different. What does this mean? Explanations should be added.
Minor
1. In the 1st paragraph of introduction, the sentence “The Ruminococcaceae, Eubacterium, and Faecalibacterium are the main bacteria producing SCFAs” is unrelated to the upstream and downstream context. It’s better to reorganize the paragraph.
2. The resolution of the figures could be improved.
3. There are some grammatic errors and inappropriate expressions in the manuscript, like “changed existed” in line 291 292, “study studied” in line 297. In line 224, “was” should be deleted. Overall, the language of the manuscript could be improved.
References
1. Ni, J. et al. Analysis of the relationship between the degree of dysbiosis in gut microbiota and prognosis at different stages of primary hepatocellular carcinoma. Front. Microbiol. 10, 1458 (2019).
2. Ren, Z. et al. Gut microbiome analysis as a tool towards targeted non-invasive biomarkers for early hepatocellular carcinoma. Gut 68, 1014–1023 (2019).
3. Zhang, L. et al. Relationship between intestinal microbial dysbiosis and primary liver cancer. Hepatobiliary Pancreat. Dis. Int. 18, 149–157 (2019).
Author Response
Hepatocellular carcinoma (HCC) is one of the most common causes of cancer death. The association between gut microbiota and various stage of HCC has been largely investigated1–3. In this work, the authors examined the differences and internal relations of gut microbiota between advanced HCC patients and healthy people.They showed Nseqs index in advanced HCC patients was significantly different compared with healthy individuals, with butyrate-producing bacteria decreased and LPS producing bacteria increased, which is consistent with previous research2. Meanwhile, they also found Lactobacillus, Anaerostipes, Fusicatenibacter, Bifidobacterium and Faecalibacterium were significantly correlated with AFP, ALT, AST and PIVKA. The research is straight-forward but the findings are not very novel, I recommend that this paper not be accepted without major revision.
There are several major and minor points that should be addressed:
Major:
- The authors should explain more in the introduction about why they focused on the differences in gut microbiota between advanced (not other stages) HCC patients and healthy people. There is already work done in this1. What is the novelty of the work?
A:Thank you for your suggestion. Liver cancer is generally found to be advanced. Among them, the specimens of patients with early and medium-term are difficult to collect. At present, patients with immunotherapy are generally used for patients with advanced liver cancer. But there is very few research on liver cancer. We have repeatedly described "The studies show that gut microbiota in tumor patients can affect the efficacy of immune checkpoint inhibitors. Therefore, understanding gut microbiota composition could guide the treatment options and evaluate the efficacy of treatment for patients with advanced hepatocarcinoma. However, in-depth research on the relationship between altered gut microbiota composition and clinical indicators in advanced hepatocarcinoma is limited. This study intends to investigate the differences in intestinal microbiota in patients with advanced hepatocarcinoma and healthy people. We wish to provide experimental basis and theoretical support for using microbiota to regulate the immunotherapy, achieve a potential biomarker for diagnosis, and improve effect of clinical treatment for patients with advanced hepatocarcinoma.” in line 49—64.
- For figure 5, there is no detailed description in the main text about the results analyzed by PICRUSt. The paper could benefit from an explanation for enzyme function and explanation for differences in enzyme abundance between patients with disease and healthy people. More explanation about Fig. 5 should be added to the main text.
A:Thank you for your suggestion. Figure 5 is a functional prediction. It may not be accurate and requires subsequent certification. Therefore, some analysis on results and content are not described in detail. We have added this idea in line 207—210, 320—323 and one reference.
In figure 2, the authors found that the Ace index and Heip index were not significant at the OTUs level, while the Nseqs index were significantly different. What does this mean? Explanations should be added.
A:Thank you for your suggestion. As with most studies, due to regional and dietary differences, clinical samples were less uniform. The Heip index represents the uniformity of the sample community, while Nseqs represents the number of sequences. These two values represent the different levels of diversity, so it does not mean that all the exponential changes will be consistent. We have added this idea in line 245—244.We have discussed in our manuscript in line 247—254.
Minor
- In the 1stparagraph of introduction, the sentence “The Ruminococcaceae, Eubacterium, and Faecalibacterium are the main bacteria producing SCFAs” is unrelated to the upstream and downstream context. It’s better to reorganize the paragraph.
A:We have deleted this sentence “The Ruminococcaceae, Eubacterium, and Faecalibacterium are the main bacteria producing SCFAs”
- The resolution of the figures could be improved.
A:I apologize for the inconvenience. We have improved the resolution of all Figures.
- There are some grammatic errors and inappropriate expressions in the manuscript, like “changed existed” in line 291 292, “study studied” in line 297. In line 224, “was” should be deleted. Overall, the language of the manuscript could be improved.
A:Thank you for your suggestion. We are so sorry for our mistakes. We have improved language of the manuscript, you can see our modification in the paper.
Round 2
Reviewer 1 Report
The authors have addressed most of the comments; they have also tried to make changes according to the reviewers’ suggestions. After revisions, the quality of the manuscript has been adequately enhanced. Therefore, the manuscript could be considered for publication in the Journal. However, there are still some editing/ syntax errors present in the manuscript which need to be corrected, hence the publishing team is advised to read the manuscript carefully before publishing.
Author Response
Many thanks for your kind reply.
Reviewer 2 Report
Thanks to the authors for their response and revisions; however, there are still some problems remaining:
1. I am still concerned about the novelty of the work. I suggest that the authors compare their work and previously published work12 and better define the significance of their work.
2. I strongly suggest that the authors explain more about the information they present in figure 5; this may help with improving the novelty of the work. And it’s better to add a list of names to the KEGG enzyme number in the supplement to make it more understood by the reader.
3. In the response, the lines the authors refer appear to be off from my version of the manuscript, so it’s hard for me to track the changes.
Reference
1. Zhang, L. et al. Relationship between intestinal microbial dysbiosis and primary liver cancer. Hepatobiliary Pancreat. Dis. Int. 18, 149–157 (2019).
2. Ni, J. et al. Analysis of the relationship between the degree of dysbiosis in gut microbiota and prognosis at different stages of primary hepatocellular carcinoma. Front. Microbiol. 10, 1458 (2019).
